# Partnerships at the Interface of Education and Mental Health Services: The Utilisation and Acceptability of the Provision of Specialist Liaison and Teacher Skills Training

**DOI:** 10.3390/ijerph20054066

**Published:** 2023-02-24

**Authors:** Mina Fazel, Emma Soneson, Elise Sellars, Gillian Butler, Alan Stein

**Affiliations:** 1Department of Psychiatry, University of Oxford, Oxford OX3 7JX, UK; 2Department of Experimental Psychology, University of Oxford, Oxford OX2 6GG, UK; 3Oxford Cognitive Therapy Centre, Warneford Hospital, Oxford OX3 7JX, UK

**Keywords:** schools, teacher education schools, mental health, mental stress, health services

## Abstract

Partnerships between school staff and mental health professionals have the potential to improve access to mental health support for students, but uncertainty remains regarding whether and how they work in practice. We report on two pilot projects aimed at understanding the implementation drivers of tailored strategies for supporting and engaging front-line school staff in student mental health. The first project provided regular, accessible mental health professionals with whom school staff could meet and discuss individual or systemic mental health concerns (a school ‘InReach’ service), and the other offered a short skills training programme on commonly used psychotherapeutic techniques (the School Mental Health Toolbox; SMHT). The findings from the activity of 15 InReach workers over 3 years and 105 individuals who attended the SMHT training demonstrate that school staff made good use of these services. The InReach workers reported more than 1200 activities in schools (notably in providing specialist advice and support, especially for anxiety and emotional difficulties), whilst most SMHT training attendees reported the utilisation of the tools (in particular, supporting better sleep and relaxation techniques). The measures of acceptability and the possible impacts of the two services were also positive. These pilot studies suggest that investment into partnerships at the interface of education and mental health services can improve the availability of mental health support to students.

## 1. Introduction

The school-based provision of mental health care is gaining momentum in many countries, with some evidence showing that young people prefer to be seen within the school context and that many are accessing mental health support primarily within their schools [1,2,3,4,5,6]. School-aged children and adolescents with emotional and behavioural difficulties are at high risk of developing later mental health problems, as well as experiencing a range of negative outcomes in their academic performance, peer and family relationships, and future occupational prospects [7]. The evidence supporting the role of early intervention for mental health problems is strong, but one of the greatest difficulties is knowing how best to design mental health support to optimise accessibility and acceptability to children and adolescents.

Schools are a key point of access for mental health promotion, prevention, and early intervention [1,8,9,10,11]. The current landscape of school-based mental health support is multi-faceted and complex. In Figure 1, we made an illustration to demonstrate some of the myriad supports and interventions that can be provided under the umbrella of ‘school-based mental health services’. The individuals providing the mental health support can range from those within the school context (e.g., peers, pastoral support workers, teachers, and school counsellors) to those external to the school but working in the school location (e.g., third-sector/charity workers and mental health professionals from statutory services). Furthermore, there are many different types of interventions that can be offered within the school setting, the targets of which range from the whole school to classes or at-risk student groups. These interventions, drawing from a variable evidence base, range from whole-school social–emotional learning approaches to cognitive–behavioural strategies and supportive counselling [12]. A number of recent systematic reviews have brought together the wealth of data that are being gathered on school-based mental health interventions—ranging from wellbeing [13,14] to interventions using either a cognitive–behavioural approach to prevent depression and anxiety [6,15] or mindfulness [16,17]. These different reviews highlight between them the breadth of interventions that have been tried, although there are few that show sustained, positive impacts on mental health. Therefore, the aims of these two projects are to focus more on teacher skill development to complement some of the learning in this area [18,19] and how to build better relationships with mental health services [20].

Across the globe, there are different initiatives being trialled, and given the different levels at which these can take place, as well as the very different school and health contexts that exist, some similar themes are emerging. Service provision is often provided from staff within schools but also from organisations outside of schools, such as charities, as well as from a range of health and allied health professionals. The evidence base does seem the strongest for interventions where a mental health professional is involved in some aspect of service delivery. Of note, many studies have come from the United States of America and Australia, making it unclear how these findings can be generalised to other places and contexts [6,12,21,22]. It is therefore important to ensure that interventions are evaluated in the contexts in which they are implemented, and we describe two such programmes here.

As indicated in the figure, there are a number of prevailing barriers and uncertainties pertaining to mental health provision in schools, including constraints related to the cost-, time-, and resource-intensive nature of many existing school-based interventions [23,24]. Another significant barrier relates to determining where the responsibility for mental health provision in schools lies within the continuum between health and education services. In addition, confidentiality, consent, and the limited evidence base remain areas that need better clarification, although they are not discussed here. These different barriers might explain why, although there has been considerable interest in developing school-based mental health interventions, the implementation of research into practice has been painstakingly slow [25].

A key area of uncertainty across schools and contexts is how best to support school staff to meet expectations for an expanded role in student mental health. Although school staff shoulder an ever-increasing number of student responsibilities—with learning processes and academic attainment often placed as priorities in their initial training—the reality remains that they must also manage student emotional and behavioural presentations. The experiences and relationships that students have with the adults in their school are therefore central to schools’ approaches to student mental health [26]. Teachers and other school staff are often some of the only adults outside of immediate family members who have a good knowledge of individual students and, therefore, can often be trusted and relied upon by students for more than their academic needs. Furthermore, school staff are likely well-placed to identify difficulties early, as well as provide early support, as students might not be able or willing to access other sources of support for a host of reasons, ranging from not wanting their families to know about their difficulties to concerns about meeting and confiding in an individual who is not familiar to them.

Student mental health is, however, not an area that has traditionally been the focus of teacher training programmes, and so many school staff feel poorly equipped to manage the emotional and behavioural difficulties that students and their families present with at school [27,28,29,30]. Indeed, many school staff perceive a need for additional training and support to help them respond appropriately and effectively to student mental health difficulties [31,32]. How to provide such support in a feasible and acceptable way, however, remains unclear. There are many factors to balance, including how best to accommodate the different and important roles that school staff fulfil for their students, work within time- and resource-limited school environments, maintain a balance between academic and mental health outcomes, and protect students’ autonomy and confidentiality. These complex dilemmas facing school staff who want to support student mental health might be exacerbated by a lack of confidence in how to manage these delicate situations [31,32].

As greater emphasis is placed on school-based mental health [11,33], as well as improving access to support [34], it is therefore important to determine what types of mental health training and support might better equip staff working in these settings to manage the range of mental health problems that present within schools. We report here on two pilot projects aimed at better understanding the implementation drivers [35] of two different tailored strategies for supporting and engaging front-line school staff in student mental health: one by providing regular accessible mental health professionals with whom to meet and discuss individual or systemic concerns (a school ‘InReach’ service) and the other by offering a short skills training programme centred around a set of commonly used psychotherapeutic techniques (the School Mental Health Toolbox; SMHT) (see Appendix A for an example of the training slides). This paper brings both these projects together to present some of the implementation considerations and practical learning points that can inform the design and delivery of mental health training and support to staff working in schools.

## 2. Project 1: Mental Health Service Liaison to School Staff—The InReach Programme

### 2.1. Intervention Description

Amongst the many innovations taking place in schools is a move to improve access to mental health support for students, through improving, for example, partnerships between education and mental health services. One example of this is the government initiative in England to enhance school provision with the training of Educational Mental Health Practitioners, whose role was designed to bridge the gap of mental health support between education and formal mental health services by providing preventive and early interventions for students with difficulties that do not necessarily meet thresholds for mental health services but raise concerns at school [36]. This study took place in the years immediately before these workers were introduced and can therefore provide insight into the range of difficulties that these practitioners might encounter.

At the time of the study, the usual mental health provision in schools was very variable and primarily determined by each school deciding whether they are able to allocate resources to this area of need. Some schools invest significantly in school counsellors or school psychologists who provide direct support to students, and other schools have relationships with charities and third-sector organisations who are usually contracted to come into the school and provide a range of different mental health support services; there remain many schools with no additional provision beyond what teachers are able to provide. There are no clear school mental health regulatory procedures, and so the practice is very school-dependent. Some schools have good relationships with statutory child and adolescent mental health services (CAMHS), but usually in the context of particular children under the care of CAMHS who need additional support at school. There have been a number of changes to the way in which CAMHS have been organised, with the more recent changes leading to working more closely with community partners [34]. An initial evaluation of the national initiative to bring ‘Mental Health Support Teams’ into schools, with Educational Mental Health Practitioners being introduced into pilot schools highlighted how these practitioners are working more with individual children than supporting the ‘whole school’ with approaches [37].

We report on the first three years (2015–2018) of a school liaison (‘InReach’) programme, whereby mental health specialists worked directly with secondary schools across one English county. The aim was to provide liaison to school staff in recognition of the high number of emotional and behavioural difficulties that these staff members manage in their usual work, with only a small proportion of these students being referred to external services. Within this programme, over 15 different mental health professionals worked with the schools over the three years of the project, all from statutory child and adolescent mental health services (CAMHS), and they are referred to here as ‘InReach workers’. They were allocated time through commissioning arrangements between mental health services and the clinical commissioning group to work for half a day each week with each specified secondary school, and they could each have between one and three schools on their caseload, alongside their usual CAMHS duties. This meant that the schools would have as consistent an InReach worker as possible in order to facilitate better relationships and knowledge of each allocated school and its needs and population. The majority of the InReach workers were staff who had a higher mental health qualification and usually at least three years of clinical experience. On average, these workers were a grade below a clinical psychologist and had either mental health nurse or mental health practitioner training. There were periods of time where CAMHS pressures or staffing vacancies led to the InReach work being less frequent, and there were also some schools in the area that did not meet regularly with their InReach worker, with work pressures and other priorities being likely contributors.

On average, the InReach workers would go at an allocated time each week to the school that they were working with. Most schools nominated a link member of staff—often the lead in the school for pastoral care, sometimes the school counsellor, and sometimes the staff working with students with special educational needs. The InReach worker would usually meet with the link member of staff each week, and these meetings became the cornerstone of much of the work that was conducted. The school climate and broader staff and student issues could be discussed so as to ensure that the work was as systemically informed as possible. During these meetings, specific issues would be discussed, or other members of school staff who would want to talk about a concern could book time for consultation with the InReach worker during their time at the school. The time was also used by the InReach worker to meet with students, either by conducting mental health literacy work, for example, by giving a school assembly or speaking to identified students about anxiety, or to assess and/or review individual students about a specific mental health presentation.

### 2.2. Participating Schools

All state-funded secondary schools in Oxfordshire, UK, were eligible and were allocated a member of CAMHS staff to be their liaison ‘InReach’ worker.

### 2.3. Outcome Assessment

As part of the study, the InReach workers were asked to complete an online log form each time that they had contact with their identified school. These logs asked the InReach workers to record (1) the types of advice and support that they were providing, (2) the areas of need that they were being asked to address, and (3) their possible impact. As the study progressed, the forms were refined to include more granular detail (for example, from 2015 to 2017, the forms asked about ‘emotional problems’ as an area of need; in 2018, they were revised to include multiple sub-categories, including anxiety, low mood, and self-harm). We report outcomes in terms of absolute and relative frequencies.

The study data from the InReach workers were collected as part of a service evaluation, and further ethical approval for additional data from school staff was granted by the University of Oxford Medical Sciences Interdivisional Research Ethics Committee (reference R67487/RE001).

### 2.4. Key Learning

The InReach workers returned a total of 1246 logs from 2015 to 2018, representing 1233 visits (all but 8 of which were in secondary schools). Across these visits, the InReach workers reported the undertaking of 1653 different activities (many recorded both main and additional activities at these visits). The most common activities undertaken according to these logs were meeting with one (N = 528 instances; recorded in 42% of all logs) or multiple school staff members (N = 410; 33%), followed by individual student (N = 265; 21%) and group case work (N = 163; 13%). Other activities included teaching/training/presentations for students (N = 74; 6%) and for staff (N = 67; 5%). Finally, the InReach workers reported N = 146 other activities across the logs (12%).

Within their school visits, the InReach workers reported discussing a range of areas of need (Table 1). Again, in many of these visits, multiple areas were discussed. The most common areas discussed were emotional difficulties in students, which were recorded in over half of the logs (N = 732 instances; recorded in 59% of logs). The more granular data collected in the 2018 logs revealed that anxiety was the single most frequently discussed area of need, comprising 62% of all emotional-related discussions and appearing in 27% of all logs from that year.

The InReach workers speculated that there might have been a number of possible impacts of their activities within schools. The most common potential impacts were giving school staff useful advice and support with strategies to help manage the student(s) discussed (endorsed in N = 580 logs; 47%) and liaison/improved communication with CAMHS (N = 395; 32%). Other potential impacts were enhanced safeguarding (N = 148; 12%), liaison with other services (N = 86; 5%), and signposting to other services (N = 43; 3%). In terms of the potential impacts on referral to CAMHS, 86 logs (7%) thought that, subsequent to the InReach activity, there was a reduced likelihood of referral, while 74 (6%) mentioned an increased likelihood.

### 2.5. Implications for Practice

The liaison service was most commonly used as a consultation and advice service for school staff members, followed by individual and group work with students.Emotional difficulties, particularly anxiety, was the most common area of need discussed with the mental health professionals.The mental health professionals perceived several potential benefits of the service, most notably in terms of providing timely advice, giving needed support in how to manage students within the school setting, assisting with safeguarding concerns, and improving communication with formal mental health services.

## 3. Project 2: Mental Health Skills Training for School Staff—The School Mental Health Toolbox (SMHT)

### 3.1. Intervention Description

The SMHT was developed in response to many frontline school staff asking to have more training in key psychotherapeutic skills to respond to student mental health difficulties. As school staff are the group most likely to be managing emotional and behavioural problems within the structured and unstructured time children spend at school, it is important that they feel confident and prepared in this role. Many of the early signs, such as disruptive behaviour, poor sleep, and difficulty drawing on existing support structures, might potentially be helped with a range of straight-forward interventions. However, whilst students can receive a range of support services at school, these are often not evidence-based [38]; therefore, the aim of this toolbox was to give school staff members additional evidence-based strategies to employ when interacting with their students, especially those for whom they have concerns.

The toolbox drew from existing cognitive–behavioural therapy (CBT) and other commonly utilised techniques and training materials and focused on areas informed by stakeholders, including teachers and pastoral support workers in schools [39]. None of the methods were newly developed, although they might have been presented in a novel manner. The front-line school staff were then trained in a two-hour session by M.F., the project principal investigator and a consultant in child and adolescent psychiatry, in how to use the ‘toolbox’ of psychological interventions with accompanying practice elements.

The SMHT approached the subject of mental health support with an open discussion under the heading of ‘ice-breakers’ to enable those attending to share how they might support an individual who has newly arrived or is not engaging with peers. This was to encourage early participation in the content of the SMHT training but also to draw on the wealth of experience and expertise that lies within any body of front-line school staff who are constantly finding creative ways to support their students. The ice-breaker activity further aimed to highlight the importance of drawing on the natural support systems within schools and harnessing peer support networks in order to maintain mental health, as well as to intervene early in emerging difficulties. In addition, the general commonly used tools of active listening and discussing a positive/pleasurable activity were practiced and discussed.

The bulk of the training focused on seven ‘tools’ [40,41,42,43]:Sleep hygiene: teaching the principles of good sleep in relation to the environment, the awareness of activities and food that can act as stimulants interfering with sleep, and the importance of maintaining a regular circadian rhythm.Behavioural change to address mood and/or anxiety difficulties: identifying and supporting behaviour changes in students who are becoming withdrawn from usual activities.Relaxation techniques: three different relaxation techniques described, namely, deep breathing, progressive muscle relaxation, and the ‘peaceful scene’.Addressing potential social isolation by better understanding students’ current networks of support using a ‘map’. This aims to help students appreciate that there can be a number of different individuals whom they may call upon when they become distressed and that they might not be as isolated as they perceive themselves to be. The map can also highlight areas where it may be useful to build additional support.A ‘treasure box’ of how students can help themselves in difficult situations. This is tailored to each specific individual and their problems, but it has the general aim of showing them that they have the capacity to help themselves when they are in distress or difficulty.How to approach students who have experienced traumatic events: this focuses on psychoeducation and learning about the importance of referral to specialist services and how to support students in that process.Problem-solving techniques (time permitting): this draws upon collaborative problem-solving approaches and follows a number of pre-defined steps with an emphasis on identifying solutions.

### 3.2. Participating Schools

Schools and local authorities were approached with an offer to receive the SMHT training. The training sessions were offered between 2015 and 2017 (1) on school premises, where all participants were from the same school, and (2) at a local lecture hall or large site, where staff from different schools participated together. In total, training was provided in eleven different locations across England and Wales. All teachers, teaching assistants, pastoral support staff, and third-sector workers supporting children within secondary schools were invited to participate in the training. Recruitment was especially targeted at those working with vulnerable populations, but all were welcome with no requirements regarding the current level or nature of contact with students.

### 3.3. Outcome Assessment

We assessed utilisation and engagement by measuring how many of the mental health tools had been used three months after the SMHT training. The participants were asked to complete an online questionnaire that asked which of the tools they had used (if any) and with how many students. In addition, they were asked to rate the helpfulness of the training (‘How helpful did you find the session?’ with a 5-point response scale from 1: ‘not helpful’ to 5: ‘very helpful’) and the likelihood that they would recommend the training to others (‘How likely is it that you would recommend the toolbox training to a friend or colleague?’ with an 11-point response scale from 0: ‘not at all likely’ to 10: ‘extremely likely’). (N.B. a series of other pre- and post-training questionnaires were collected, including self-efficacy and social desirability measures, to be reported elsewhere.) We report outcomes in terms of means, standard deviations, and absolute and relative frequencies.

The study was approved by the NHS Social Care Research Ethics Committee (reference 15/IEC08/0055).

### 3.4. Key Learning

In total, 223 attended the training, of whom 192 provided details of their jobs. Of these, teaching staff made up the largest group of participants at 52 (27%) teachers and 30 (16%) teaching assistants. Other participants included counsellors working in third-sector organisations providing support in schools and voluntary workers. A total of 105 training attendees completed the post-training questionnaires (49% of those who completed the pre-training questionnaires). Of these, 89 (85%) reported using at least one of the tools. Table 2 shows the proportion of all respondents (i.e., N = 105) who reported using the different tools in the first three months following the training. The relaxation techniques and sleep advice were the most commonly used tools, with 44 respondents (42%) reporting having used each of these. The next most commonly used tools were active listening (N = 37; 35%) and the treasure box (N = 33; 31%), each used by approximately one-third of respondents.

Of the 105 respondents, around half (N = 50; 48%) reported having used the tools with students with emotional and behavioural difficulties. The tools were less commonly used with other groups, including students with learning difficulties (N = 14; 13%) and newly arrived students (N = 13; 12%). Thirty-five respondents (33%) reported using the tools with students outside these categories. In terms of the number of students reached, of the 92 respondents who answered this question, 85 (92%) reported having used the tools with at least one student. A total of 16 (17% of total respondents for this question) reported having used the tool(s) with one student, 22 (24%) with two students, 18 (20%) with three students, 9 (10%) with four students, and 20 (22%) with five or more students. Finally, 24 respondents (23%) reported that they had used the tools on themselves. There was no clear pattern to identify which participants were most likely to use the tools. For those who had used the tools five or more times, we had data on 13 of the 20 individuals, and they represented teachers (n = 8), teaching assistants (2), third-sector volunteers (2), and an educational psychologist (1).

In terms of the acceptability of the training, most respondents found the training helpful and would recommend it to a colleague. On average, the 105 respondents found the training helpful (on a scale of 1 to 5, with 5 being ‘very helpful’, mean = 4.2, standard deviation = 0.9), with half (N = 53; 50%) rating the training as ‘very helpful’ (the highest level), and no respondents rating it as ‘not helpful’ (the lowest level). Most would recommend the training to a colleague, as the mean rating for the 101 respondents who answered this question was 7.7 (on a scale of 0 to 10, with 10 being extremely likely; standard deviation = 2.0).

### 3.5. Implications for Practice

Three months after participating in a mental health skills training programme, 85% reported having used the skills learned during the training with a student.The most ‘popular’ skills in terms of post-training use were the relaxation techniques and sleep advice.Skills were most commonly used to support students with emotional and behavioural difficulties, though many staff also found them useful for their own mental health and wellbeing.Nearly all staff found the training at least moderately helpful, and most would recommend it to their colleagues.

## 4. Discussion

### 4.1. Main Findings

There is growing emphasis on the role of school staff in supporting student mental health, yet many report not having the needed skills or confidence. This paper reported on two pilot studies that explored the implementation drivers of partnership working initiatives between education and mental health specialists in the United Kingdom. Overall, the findings demonstrate that such partnerships have potential value for supporting school staff to respond to student mental health concerns and suggest that further investment in this area is merited.

The first study aimed to explore the utilisation and possible impacts of a county-wide school InReach programme of specialist support and advice for staff within the school setting. The InReach pilot study found that mental health professionals working in CAMHS can play an important role in supporting school mental health interventions, as evidenced by the many and varied activities recorded. Most commonly, they provided consultation and advice to staff members or provided support to individual students or groups of students, particularly around anxiety and emotional difficulties. The InReach workers perceived many possible impacts of their partnership, most notably in terms of offering specialist advice and improving communication with CAMHS.

The second study aimed to assess the utilisation and acceptability of the School Mental Health Toolbox (SMHT), a skills-based mental health training programme for school staff. The findings indicate that the training equipped school staff with a range of techniques and strategies that they could use with their students. The uptake of the tools learned in the training was high: nearly all staff members who completed the post-training survey had used at least one of the tools in the three months following the training. The most ‘popular’ tools were the relaxation techniques and sleep advice, and staff members reported using the tools predominantly with students with emotional and behavioural difficulties. In terms of the acceptability of the training, nearly all staff found it at least moderately helpful, and the majority indicated that they would recommend it to a colleague or friend.

### 4.2. Partnership Working—A Feasible Solution for a Well-Documented Need

Changing professional behaviour is a complex task, and although many different mental health training programmes are offered to education services across the country, it is unclear which is the best mode to deliver such training and whether skills from the training are ever then utilised. There is a large body of literature on school-based mental health interventions and approaches for treating mental health problems, yet most of these evidence-based interventions are delivered by psychologists, counsellors, or other mental health professionals [12,24,44,45,46]. Few have successfully utilised school staff, and there are limited data investigating long-term implementation and impact.

These two projects are potentially scalable ways to harness the goodwill between services to better work together to address the needs of students within schools. Furthermore, as many of the drivers of emotional difficulties—particularly anxiety—can be fuelled by the school experience (from both interpersonal and achievement stressors), it is relevant that much of the discussion with mental health professionals and the toolbox utilisation of sleep support and relaxation techniques were potentially addressing these perceived student difficulties. This is aligned with the findings of another example of the impact of improved partnership working between education and mental health services that was observed in the UK TaMHS (targeted mental health in schools) project, which demonstrated how stronger links with local mental health services improved behavioural management in primary schools and that most parents accessed schools as their first point of call for mental health difficulties in their children [47]. Evaluations are in process of the English Educational Mental Health Practitioner initiative, providing a range of support services in schools [37,48].

### 4.3. Knowledge Gaps and Future Directions

Although these pilot studies offer a wealth of information regarding the real-world utilisation and acceptability of partnership working strategies, there remain a number of unanswered questions to be explored moving forward in terms of the effectiveness and implementation of such strategies. Here, we present several of these:*How do school staff perceive mental health liaison services?* Although we had originally planned to interview school staff regarding their experiences of the InReach programme, this was not possible due to the disruptions of the first lockdown of the COVID-19 pandemic. Whilst the InReach workers perceived many possible benefits of their work, it is critical to know how the service was viewed by school staff.*What were school staff’s experiences of using the SMHT tools, and what were students’ experiences of them?* Understanding the implementation drivers associated with the training goes beyond whether school staff used the tools; in addition, it is important to understand the perceived and actual mental health impact of the tools from the point of view of those delivering them (school staff) and receiving them (students).*What do CAMHS workers and school staff members perceive as the barriers and facilitators of partnership working?* Further qualitative research can help better understand how best to design, implement, and evaluate the impact of partnership working.*Does partnership working improve school staff members’ confidence and preparedness for addressing student mental health difficulties?* As the main aim of these services was to support school staff, it is important to understand the impact on staff attitudes and self-efficacy towards student mental health. Gathering data on the existing skills of the staff prior to training would help to understand any differential impacts of additional training.*Is partnership working associated with improved mental health outcomes for students?* As we designed these studies as pilot projects, we did not include any mental health outcomes for students. Well-designed, adequately powered studies are needed to understand the effectiveness and cost effectiveness of these strategies.*Is partnership working associated with improved school climate and home environment?* It would be important to find ways to measure the broader systemic impacts of work conducted in schools and how this might impact the general experience of children within both the classroom and other school activities, as well as if there are impacts on families, including on parents and siblings.

## 5. Conclusions

The school experience has the potential to both enhance and threaten the mental health of students. As such, gaining a better understanding of whether and which school-based support services might be of value to students and staff is a priority. The pilot studies presented in this paper suggest that additional investment into partnerships at the interface of education and mental health services may be useful for improving child and adolescent mental health support. Working through school staff is potentially sustainable and scalable, as they already spend so much time with students. Many staff members have relationships with students that are more meaningful than those that other extra-familial adults might have with them. Furthermore, conversations between staff and students can take place in a location that can be accessible to students and assist school staff. Ensuring that the many innovations and initiatives in this area follow a framework where their implementation is evaluated will hopefully ensure that learning about how to improve mental health support within educational settings is maximised and, thus, reaches more students.

## Figures and Tables

**Figure 1 ijerph-20-04066-f001:**
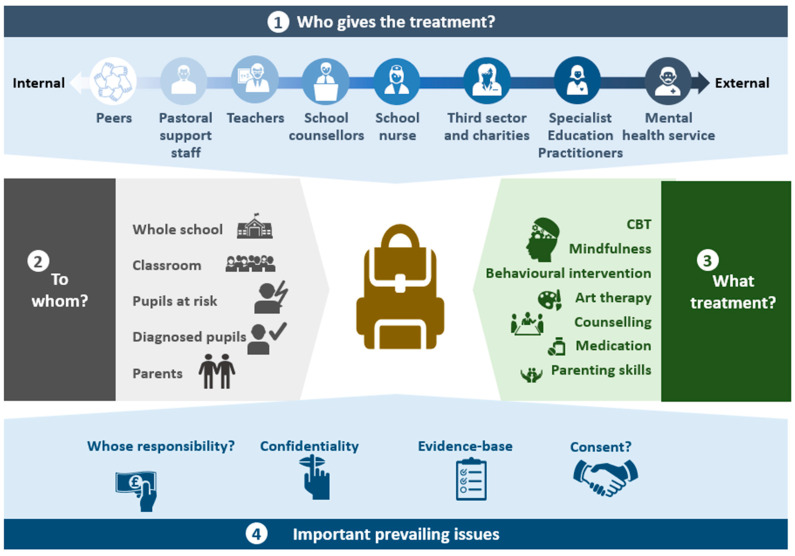
Mapping school-based mental health interventions.

**Table 1 ijerph-20-04066-t001:** Areas of need discussed between mental health professional ‘InReach’ workers and school staff.

Area of Need in Student/s	2015–2017 Responses	2018 Responses	Total for Each Area of Need (% of Logs in Which the Area Was Recorded)
Emotional difficulties	369	363 (anxiety 224; low mood 70; self-harm 69)	732 (59%)
Behavioural problems	232	82 (anger and aggressive behaviour 62; bullying 20)	314 (25%)
Planning with the school	121	154	275 (22%)
School management	138	91 (broader school management issues 70; safeguarding 21)	229 (18%)
Social difficulties	159	49 (peer problems)	208 (17%)
Parenting issues	74	79	153 (12%)
Total	1093	818	1911

%s sum to >100% because each log could have >1 area of need discussed. Additional areas were included in 2018–2019 responses.

**Table 2 ijerph-20-04066-t002:** SMHT tool use 3 months post-training.

SMHT Tool	Number of Responses (% of Respondents)
Relaxation techniques	44 (42)
Sleep hygiene advice	44 (42)
Active listening	37 (35)
Treasure box to prepare for difficult situations	33 (31)
Ice-breakers	20 (19)
Problem solving	20 (19)
Behavioural change	16 (15)
Pleasurable experiences	15 (14)
Map of social networks	13 (12)
How to help a student who has experienced trauma	12 (11)

## Data Availability

Project 1: The documentation of the liaison meetings with school staff are not anonymised and therefore not publicly available. The study authors are able to investigate specific questions upon reasonable request. Project 2. Anonymised qualitative data interview transcripts are available from the study authors. Other data are available from the authors upon reasonable request. The SMHT (toolbox) is available for anyone to download and use and is included in Appendix A.

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
