# Peer review of "Partnerships at the Interface of Education and Mental Health Services: The Utilisation and Acceptability of the Provision of Specialist Liaison and Teacher Skills Training"

_ijerph, 2023, doi:10.3390/ijerph20054066_

Round 1

Reviewer 1 Report

Thank you for the opportunity to review your manuscript. I found the topic to be interesting and timely, especially since the data was collected prior to COVID-19 and would hav insight for the growing mental health needs of students and school professionals. The concept is interesting and has the potential to impact the field. However, the data is lacking on the actual study. More information is needed about the participants in both projects, the mental health workers in project 1, and the trainers in project 2. Also, more information is needed about the methods-data collection process, data analysis, etc. 

Project 1:

I found the concept to be interesting and one that has the potential to impact the field. However, I was left wanting more information to fully understand the process, what benefits it had if any, and how schools could replicate this model.

On page 4 you briefly provide some background about the 15 mental health professionals working with the schools. I would recommend adding in more specific demographic data about those workers, such as highest degree earned, current job title, etc. to get a better picture of their expertise as a mental health consultant for the schools. I would recommend including more information about the process. For example, how did schools meet with a worker, how did the meeting stake place, what types of people within the schools met with the workers. I recommend adding more information to the methods section for this portion of the paper to better understand the participants, data collection process, and how the data was analyzed. You have a good rationale for the study and concept for the two projects. More information is needed to back up the claims and implications for practice. 

Project 2:

You did a nice job of describing the intervention and the seven "tools". I encourage you to add more specific data about the 105 participants (ex. highest degree earned, job titles of each, which districts/schools had which representatives, etc.). 

For the key learning section you do a nice job of including a chart. I would be curious if the findings were analyzed based on job title. For example, did school psychologists use more of the strategies from the training than secondary educators? Does the job background of the participant impact how the training was used?

Author Response

Thank you for your review, we attach our response to Reviewer comments here

Reviewer 2 Report

The manuscript addresses an issue of great relevance today, given the increase in mental health problems among the school population. It should be considered that these problems have been notably exacerbated as a consequence of the recent pandemic, the temporary closure of educational centers and confinement. In this situation, coordinated action between mental health services and the school environment may be a particularly useful strategy. For this reason, the manuscript reports two experiences aimed at promoting this alliance. Nevertheless, some aspects need to be reviewed:

-A more appropriate selection of keywords is proposed to the authors. In this sense, authors are suggested to avoid using the same words in the title and in the keywords. For the search of the latter, it is recommended to use the UNESCO Thesaurus database.

-The number of sources consulted (38) is somewhat low, so the authors are actively encouraged to broaden the bibliographic search. This extension should be used to incorporate more updated sources, since only 18 of the 38 have been published in the last 5 years.

-It would be advisable to add the source from which Figure 1 has been extracted or, if it is own elaboration, to indicate it.

-On the other hand, it would be convenient to adapt the references to the standards established by the journal.

-Before describing both interventions, it would be advisable to describe the existing mental health system for the school population in the United Kingdom. This would allow the reader who is not familiar with it to get a better idea of the reality under study (what are the existing services and how do they work?).

-In terms of content, the authors are encouraged to describe in more detail the two interventions developed. It would be very interesting if, for example, they could provide additional information on the training sessions held, the duration, the resources employed, the methodology used, etc.

-In the Discussion section, it would be interesting to compare the results of both interventions with other initiatives previously developed in this area. For this purpose, it would be advisable to describe these in the first part of the manuscript as well. How are these types of needs addressed in other countries, and is coordinated action between mental health services and the school environment employed? This analysis would further enhance the scope of the study.

-It would also have been interesting to analyze the impact of this type of intervention on the family environment: were families able to see improvements in their children’s mental health? It is therefore suggested to add this aspect as one of the prospective lines of research.

Round 2

Reviewer 2 Report

The authors of the manuscript have made all the necessary changes. Therefore, the paper is publishable.